# On Rational Choice and the Representation of Decision Problems

**Paulo Oliva** [1] **and Philipp Zahn** [2,*]

1. School of Electronic Engineering and Computer Science, Queen Mary University of London, London E1 4NS, UK; p.oliva@qmul.ac.uk
2. Department of Economics, University of St. Gallen, Bodanstrasse 6, 9000 St. Gallen, Switzerland
* Correspondence: philipp.zahn@unisg.ch

**Abstract:** In economic theory, an agent chooses from available alternatives—modeled as a *set*. In decisions in the field or in the lab, however, agents do not have access to the set of alternatives at once. Instead, alternatives are represented by the outside world in a structured way. Online search results are *lists* of items, wine menus are often *lists of lists* (grouped by type or country), and online shopping often involves filtering items which can be viewed as navigating a *tree*. Representations constrain how an agent can choose. At the same time, an agent can also leverage representations when choosing, simplifying their choice process. For instance, in the case of a list he or she can use the order in which alternatives are represented to make their choice. In this paper, we model representations and decision procedures operating on them. We show that choice procedures are related to classical choice functions by a canonical mapping. Using this mapping, we can ask whether properties of choice functions can be lifted onto the choice procedures which induce them. We focus on the obvious benchmark: rational choice. We fully characterize choice procedures which can be rationalized by a strict preference relation for *general* representations including lists, list of lists, trees and others. Our framework can thereby be used as the basis for new tests of rational behavior. Classical choice theory operates on very limited information, typically budgets or menus and final choices. This is in stark contrast to the vast amount of data that specifically web companies collect about their users' choice process. Our framework offers a way to integrate such data into economic choice models.

**Keywords:** rational choice; search; path-independence; procedure; representations

## 1. Introduction

Suppose you want to buy a product from an online retailer. A search on their website will return a page containing a list of 10 alternatives, plus the option to explore the next page containing 10 more alternatives, and so on. Before you can explore item 11, you need to navigate to the next page. Item 21 will be two clicks away from your initial search.

How does the presentation of alternatives affect your decision process? How can we reason about such information and its effect? In the standard economic treatment such information does not play any role. In fact, cannot play any role: There is no obvious way how to incorporate non-choice data and information about the environment into the classical choice framework. At the same time, such data has informational value and economic relevance. Many online platforms track extensive data about their customers beyond mere choices in order to analyze their decision-making. (Researchers also increasingly study consumers' search behavior at a granular level (e.g., [1–3]) and aim to integrate different types of information in learning customers' preferences (e.g., [4])).

The question is, how do we formally model such richer choice environments? In this paper, we propose such a framework in which we model *representations* like the website above, *choice procedures* operating on them, and an *extension map* which relates procedures to

the classical choice approach in a canonical way. We conceive of a procedure as a program: Given the representation of alternatives it describes "how" an agent navigates through the representation and how he or she arrives at their choice. Our framework can describe rich choice environments and the behavior of agents interacting with this environment. It therefore brings formal choice theory and actual observable information of choice behavior closer together.

For a program we can also ask "what" it computes. It can be mapped into its input-output behavior—for each input we observe some output. (In theoretical computer science, the formal description of a program's behavior is often referred to as its *denotational semantics*, compare e.g., [5]). We can do the same for a choice procedure. The extension map relates each choice procedure to an input-output function. As it happens, in the context of choice procedures these input-output functions are well known: they are "choice functions" that output a choice for each decision problem an agent faces. Hence, there is a natural way to hide the richer description and extract the economic bare-bone information—if one wants to.

The formal mapping between choice procedure and actual choice also introduces new ways of analyzing choice. We can investigate whether and how properties of choice behavior are related to properties of choice procedures. In other words, in which way is "what" an agent chooses related to "how" he or she chooses. In economic choice theory, a central question concerns rationalizability: Can choices observed by an agent be the outcome of the maximization of a preference relation? (For an introduction to this question see, for instance [6]. There is a rich literature on different forms of rationalizability. For an overview see [7]). Thus, we begin our investigation with rational choice as the obvious benchmark: What conditions do representations and choice procedures have to fulfill so that choices can be rationalized? We distinguish two scenarios. First, we only impose minimal assumptions regarding representations: we assume that the agent only knows the shape of a representation. For instance, an agent knows that alternatives are organized in a tree but does not know which alternative is positioned where in the tree. We show, an agent with a strict preference relation must have a procedure which (i) ignores *any* representation and (ii) proceeds by a 'divide-and-conquer' strategy. The latter means that the overall choice will be determined by their choice on 'sub-problems'. Where do these sub-problems come from? They are determined by the representation! In the initial example above, an agent proceeds by divide-and-conquer if their ultimate choice is either among the choice on the first ten items or is the same as the choice on the remaining result pages.

The result that a procedure has to follow the principle of "divide-and-conquer" mirrors Plott's "path-independence" property [8] and its role in rationalizable choice functions. While these notions appear similar, and may be obvious in hindsight, there is still a subtle but important difference. Path-independence is a condition imposed on choice *behavior*. That is, if we observe a sequence of choices, we can (in principle) tell whether this condition is violated or not.

In contrast, our property operates on the level of the procedure, which is an operational description of how the agent will choose. As a consequence, if an agent describes how he or she will proceed, we know whether this condition is fulfilled before a single choice is made. This shift of perspective and its underlying methodology originates in computer science where an analogous problem exists. Computer scientists are interested in the relationship between program description and their behavior ("semantics") as they would like to understand the behavior of the programs even *before* running them. Such understanding contributes to the design of programs that behave as intended. The alternative to this is *testing*, i.e., running the programs on a range of inputs and observing how they behave.

In decision theory, when we think about rationalizable choices, in effect we are thinking about testing. Given budgets, we observe choices. If these choices are consistent, the agent chose rationally. Here, we are lifting the rationality criterion onto the "program", the agent's decision procedure, and the representation of alternatives. We ask, what structure must programs fulfill so that from its description we can tell whether choices are rational-

izable. One might be tempted to view this latter point as a technicality but we posit that having a way to express decision procedures as well as being able to check properties of their induced behavior will become the more relevant the more software systems make decisions on behalf of humans. When entrusting software with making consumption decisions, we want to make sure that the software makes good decisions *before* we let it loose.

Assuming that *nothing* is known about representations except their structure is an important edge case but arguable not the most interesting one. Therefore, we turn to a second, more realistic scenario where information about the positioning of specific alternatives in a representation is known.

We show how such additional information can be modelled in our setting and how it affects the possibility to rationalize choice. If the environment carries relevant information, rationalizable choice procedures exist which are merely operating on the shape of a representation.

To make it more concrete: Consider an agent buying a mattress online. After filtering the right size, an agent is presented with a list of alternatives. The order in the list is not random but follows some criterion, say mattresses ordered from hard to soft. If the agent's preferences are aligned with this criterion, he or she can just choose the first (or the last) element in the list—even without inspecting this alternative before.

However, we can consider this aspect also from a different angle. Online environments are designed with intentions about the choice process of customers. We can ask how a choice environment should be structured such that agents can replace a complicated procedure with a considerably simpler one that gives them the same choice. Our framework allows to pin point under which condition a given representation helps the agent to use a procedure which does not inspect all alternatives. (Naturally, this is neither the only concern in the design of such environments nor will such environments be designed for rational consumers only. However, our framework helps to understand how an environment may help to reduce frictions on the consumer end. Furthermore, companies which offer a simple and effective choice environment may have a competitive edge. In [9], for instance, many customers searching for books online exhibit a preference for a platform that cannot be explained by price differentials).

While in this paper we focus our characterization of procedures on rational choice, our framework can be applied to other decision criteria different from maximization of a preference relation. Consider satisficing [10]: an agent distinguishes alternatives by either good-enough or not good-enough. If alternatives are organized in a list, the agent will search through the list and stop as soon as he or she finds a satisfiable element. Now suppose alternatives are organized in a tree and properties of the alternatives determine their position in the tree. For instance, in the case of a mattress this could be size, spring-vs. foam-core etc. If the agent's set of satisfiable elements share this property, he or she can filter alternatives (i.e., go along the branches of the tree) and just choose any element from the remaining alternatives. The framework we propose can accommodate such alternatives to rational choice. We illustrate this for satisficing.

The paper is organized as follows. In the next section we relate our paper to the literature. The two sections after that then set the scene: In Section 3, we introduce the notion of a represented decision problem, and relate it to classical decision problems. In Section 4, we describe decision procedures on these represented problems, and relate it to choice functions. We then come to the first result: In Section 5, we introduce properties of decision procedures and use them to characterize rationalizable procedures. The second result follows in Section 6 where we consider additional restrictions of representations so that rational agents can rely on them when choosing. Section 7 concludes.

The challenge in this paper is setting up the framework, i.e., adequate definitions. The main theorems then fall out in a straightforward way. However, we consider this a feature and not a bug. As the proofs proceed in a standard fashion, we have delegated them to the Appendix B.

## 2. Related Literature

The general question how the information about the environment can influence choice is, of course, not a novel idea. The role of the external environment and its interaction with internal choice processes was already discussed in [10]. (For a more recent account see [11]). In this paper, we propose a way to model the interaction between external information and internal choice processes and analyze their properties.

Our paper is related to several recent strands of the literature. The paper closest to ours is [12] who also investigate the role of extra information on choice behavior. They provide a model which extends a classical choice problem to a tuple containing a budget and "frame". They then investigate how choices by agents who make use of the frames can be related to rational choice. There are two differences to our paper: First, instead of assuming "(...) that the frame affects choice only as a result of procedural or psychological factors" so that "additional information that is in fact relevant in the rational assessment of alternatives thus should not be regarded as frame" as in [12], p. 1288, we focus on the opposite: how can the extra information support rational decision-making. Secondly, our approach differs. In [12] properties and characterizations are in terms of behavior—analogously to classical choice theory. In our framework, properties and characterizations are in terms of the description of how agents choose. This changes the interpretation of properties and how they can be used.

Apart from [12] there are several other papers which consider specialized representations and then investigate choice functions focused on them. For instance, ref. [13] investigate procedures which operate on *lists* of alternatives. (For other papers investigating lists see [14,15]. Ref. [16] considers *tree-based* representations). In contrast, we consider representations in general and our results hold not only for specific representations. We thereby can generalize their results.

In this paper, we focus on the interaction between representations and choice procedures which result in rational choice. However, representations can be used in general to simplify procedures which in the end implement complex goals—rational or not. This wider perspective of considering the interaction between representations and choice procedures, for which we provide a general model, links our paper to two further strands of the literature. First, several papers consider sequential choice procedures where agents reduce the overall set of alternatives in steps towards a final choice. For instance, [17] show that a heuristic, choosing by checklist, where agents face a decision problem and sequentially reduce the set of alternatives by throwing out alternatives which do not satisfy desirable properties, can lead to rational choices. Representations of alternatives can simplify this approach: If alternatives are organized in a binary tree structure reflecting characteristics of the goods, then the agent can use this information to implement their checklist. Other sequential procedures which could be supported by representations are analyzed in [18–20]. In all these cases, equipped with a suitable representation, agents can operate simple procedures which will mimic their internal choice processes.

Second, standard theory assumes that agents have access to all alternatives at once. However, in practice agents often do have to search for alternatives. Starting with [21] there is an old literature investigating the consequences of search costs, typically focused on finding the best price. It is obvious how a representation of alternatives will facilitate search—just consider alternatives arranged in a list ordered by increasing prices.

More recently, search has been investigated from the perspective of bounded rationality and behavioral economics. For instance, ref. [22] study the consequences of rational inattention on agents. They derive agents' optimal "consideration sets", i.e., the set of alternatives that agents will consider at all when making a choice. Consideration sets are also analyzed in [23]. They provide a model of how an agent's consideration set dynamically evolves. Again, it is obvious that the representation of goods may be interwoven with agents' consideration sets (as [23] also point out). This could be because of the arrangement of goods by quality (average recommendations) or popularity (most frequently bought). Representations also matter when it comes to understanding how consumers navigate the

vast number of varieties they nowadays face. In [24] consumers will inspect alternatives *only if* they are members of their consideration set. Representations may help to quickly reduce the initial very large set of alternatives. (If representations are well designed, then a shop can also offer a larger variety of alternatives without overwhelming consumers). Once sufficiently reduced, agents maximize rationally. Ref. [24] provide an example of such a procedure, "Narrowing Down", where the agent reduces alternatives by repeatedly refining his search up to a point where the number of listed alternatives is below a threshold.

## 3. Representing Decision Problems

Classically, a decision problem is modeled as a subset $A$ of a set of alternatives $X$, i.e., $A \subseteq X$ and the choice function $c \colon \mathcal{P}(X) \to X$ as a function operating on the decision problem, using $\mathcal{P}(X)$ to denote the power-set (We will always assume that $A \subseteq X$ is non-empty). of $X$ (see Chapter 2 in [25] or Chapter 1 in [26]). This characterizes the behavior of an agent.

The modeling of a decision problem as a *set* abstracts from how the problem is presented (see [6], p. 24 for a discussion). The *choice function* models an agent's *behavior* mapping budgets into choices. It thus abstracts from the actual process of choosing. Furthermore, indeed the behavior of different decision-making models can be captured through choice functions. For instance, the behavior of the satisificing procedure can be modelled in this way (see Chapter 3 in [6] for a discussion).

In practice, however, agents do not see the set of alternatives (at once), but rather a representation of that set (cf. [12,13] and the related research discussed in Section 2). Consequently, the process of choosing involves navigating this represented problem. For instance, some online retailers only offer a single alternative at a time. To access more options, agents have to navigate to the next item. Or some online shops will first ask specific questions one at a time, and depending on the answers they will offer a single alternative. In this section, we propose a model for these "access restrictions", by introducing the formal notion of a representation space.

### 3.1. Representation Spaces

Before we proceed with the formal definition of a *problem representation* and of a *representation space*, let us outline two properties representations should have:

- *Representations should be inductively defined*: We want our problem representations to be *inductively defined*, i.e., the representation of large decision problems should be built from the representation of some of its sub-problems. For instance, a large list might be formed by concatenating two smaller lists, or a decision tree should be built from smaller decision trees. This is an important feature, as it ensures that represented problems are constructed in a *modular* way.
- *Representations should be parametric on the set of alternatives $X$*: Similar to the definition of a decision problem which is valid for different kinds of alternatives we also want to define representations without being tied up to a particular set of alternatives $X$. For instance, we should be able to define the "list representation" of $X$, without referring to anything specific about $X$, so that we can then deal with "lists of cars" or "lists of wines" in a uniform way. This is also an important feature, as it ensures that representations are constructed in a *uniform* way.

Recall, in the introduction we used the example of shopping online. Products $X$ are displayed as a paginated list of results, with 10 results on each page. This representation is *inductively constructed* from the set $X$ via a limited number of operations: creating a single item, organizing 10 items in a page, and finally combining all pages into a list. Moreover, the representation structure is also *uniform* in $X$, we can use the same structure for different sets of alternative $X$. Like mathematical expressions built from numbers via mathematical operations, or like sentences built from words which are themselves built from letters, search results are built from primitive objects and operations defined on them.

(In mathematics, such structured sets are *algebras*. They have led to the development of *algebraic types* in computer science on which we base our approach. See Chapters on Finite Data Types (Product Types and Sum Types) in [27]).

In other words, we can think of our search results as a formal language that uses the values of $X$ in some fixed way to build lists of pages of items. Formally, this boils down to thinking of represented decision problems as *words in a grammar*. For instance, the grammar that describes a search result in our online retailer would be:

Item $\Rightarrow x$, for each $x \in X$
Page $\Rightarrow$ [Item$_1$, ..., Item$_{10}$]
List $\Rightarrow$ Page  or  Page, List

The first line says that any $x \in X$ is considered a "Search Item". The second line says that a list of 10 items forms a "Page". Furthermore, the last line says that a "List of Results" is inductively defined as either a single page, or an initial page followed by more pages. It is this last recursive definition (we are defining List in terms of List) that allows us to describe lists of arbitrary (and possibly infinite) length in a finite way.

The above representation of a search result fulfills our two desiderata—it is both *inductively defined* and *parametric on the set of alternatives X*. Moreover, notice that any concrete search result can be presented diagrammatically as shown in Figure 1, where the various items are placed at the terminal nodes, and the internal nodes are used to describe the various labels such as "Page" or "List". A user choosing one of the items in a search result needs to navigate this structure in order to reach each of the items available.

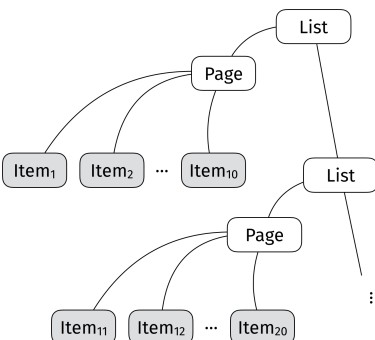

**Figure 1.** Diagrammatic representation of search result.

Let us move on to a formal definition:

**Definition 1** (*T*-representation for $2^X$)**.** *Given a set of triples $T = \{\langle C_i, n_i, m_i \rangle\}_{i \in I}$, where each $C_i$ is a name and $n_i, m_i \geq 0$ are natural numbers, with $C_i \neq C_j$ for $i \neq j$, let TX denote the set obtained by repeated application of the following rule*

*If $\{x_1, \ldots, x_{n_i}\} \subseteq X$ and $\{a_1, \ldots, a_{m_i}\} \subseteq TX$ then $C_i \, x_1 \ldots x_{n_i} \, a_1 \ldots a_{m_i} \in TX$, for each $i \in I$.*

*We call TX the T-representation space of $2^X$, and each $a \in TX$ a T-represented decision problem. We also refer to each name $C_i$ as a* constructor*, as these are the only possible ways of creating represented problems in TX.*

When constructing new elements of $TX$, we can use both, elements of $X$ (the $x_1, \ldots, x_{n_i}$) or previously constructed elements of $TX$ (the $a_1, \ldots, a_{m_i}$). So $n_i$ and $m_i$ specify the arity of the constructor $C_i$ on each of these.

**Definition 2.** *We must have at least one $C_i$ with $n_i > 0$ and $m_i = 0$, to ensure that TX is not an empty set. We will assume this is the case from now on.*

Note, the diagram above might suggest that we are restricted to representations of decision problems as trees. However, the notion of a T-representation is more general and

includes other representations of alternatives such as lists, queues, stacks, etc. (One can think of a T-representation as *sentences in a language*. In any language which is described by a grammar (which includes both computer programs but also natural language) one can "parse" the sentences of that language into a syntax tree. However, that syntax tree representation of sentences is in no way restricting the expressive power of languages themselves. Once we fix a T-representation we are in some sense fixing the grammar of the language, and the diagrammatic representation of the elements in this language is nothing more than the syntax tree of that element of the language).

Let us illustrate the definition (In functional programming these *T*-representations are known as *algebraic data types* (Chapter 8, [28]). For instance, $T_{\mathsf{List}} = \{\langle \mathsf{S}, 1, 0 \rangle, \langle \mathsf{C}, 1, 1 \rangle\}$ corresponds to the data type of non-empty lists). above using our search result example. In this case, we are actually using three nested representations. First, single items need to be represented, which we do with $T_{\mathsf{Item}} = \{\langle \mathsf{I}, 1, 0 \rangle\}$. Here we are using the name I to label an item, $n = 1$ (we use one element of $X$) but $m = 0$ (we do not use previously created items). That only allows us to represent singleton sets. For instance, if $X = \{x, y, z\}$, then $T_{\mathsf{Item}} X$ only has three elements

$$T_{\mathsf{Item}} X = \{\mathsf{I}\, x, \mathsf{I}\, y, \mathsf{I}\, z\}$$

We think of $\mathsf{I}\, x$ as representing the set $\{x\}$ as an "item" of a search result. So, the representation space $T_{\mathsf{Item}} X$ allows us to represent the three singleton sets $\{x\}$, $\{y\}$ and $\{z\}$.

However, we can then group items into a page—we think of a page as a list of 10 represented items. Formally, we would take $T_{\mathsf{Page}} = \{\langle \mathsf{P}, 10, 0 \rangle\}$. Here we use the name P to label a page, $n = 10$ (we use 10 elements to create a page) but $m = 0$ (we do not use previously created pages to create a new page). Hence, $T_{\mathsf{Item}} Y$ is the set of "pages" each containing 10 elements from the set $Y$. If we take the set $Y = T_{\mathsf{Item}} X$, i.e., items from $X$, we are then nesting representations: a page represents a list of 10 represented items. For example, given $x_1, \ldots, x_{10} \in X$, one possible element of $T_{\mathsf{Page}}(T_{\mathsf{Item}} X)$ is

$$\mathsf{P}\, (\mathsf{I}\, x_1) \ldots (\mathsf{I}\, x_{10})$$

which we view as a representation of the set $\{x_1, \ldots, x_{10}\}$. We are not ruling out repetitions in the represented problem. So we could have that all $x_i$ are equal to some $x$, which means that this page with apparently 10 items actually represents the singleton set $\{x\}$. Furthermore, note that the order in which the elements appear in the representation matters. The representation where the elements are listed in inverse order

$$\mathsf{P}\, (\mathsf{I}\, x_{10}) \ldots (\mathsf{I}\, x_1)$$

is a different representation of the same set $\{x_1, \ldots, x_{10}\}$.

Finally, the representation of the ultimate search result as a list of pages involves a choice: a list is either a single page, or a page followed by other pages. This is formally achieved by defining a representation space with two constructors $T_{\mathsf{List}} = \{\langle \mathsf{S}, 1, 0 \rangle, \langle \mathsf{C}, 1, 1 \rangle\}$ for singleton lists and compound lists. Our search result over a set of alternatives $X$ is then represented as an element of $T_{\mathsf{List}}(T_{\mathsf{Page}}(T_{\mathsf{Item}} X))$, i.e., a list of pages that each contain 10 items.

In general, a representation models how an agent can access information regarding alternatives in a fine-grained manner. Which representation makes sense, depends on the situation.

Obviously, it would be a rather futile exercise to fix a specific "grammar" such as the example above and characterize when a procedure operating on it can be rationalized. Instead, we consider grammars in the abstract. The properties we will later introduce in order to characterize a decision procedure, hold for grammars in general and not just specific cases.

Our framework is sufficiently expressive so that we could study procedures which do not pick elements of $X$ but instead represented problems such as a list of alternatives, while

this is certainly interesting, for the beginning we believe it is more relevant to understand when a procedure results in a concrete element.

Lastly, note, as is standard in the literature, we assume throughout that the set of alternatives $X$ is known. (As a consequence, the agents face no uncertainty regarding the alternatives they choose. We keep this assumption intact in order to make our results comparable to the standard in the revealed preference literature. However, in principle, the case where the agent faces uncertainty is of course interesting and could be modelled within our framework).

### 3.2. The Extension and Representation Maps

What is the relationship between represented decision problems and classical decision problems?

As representation spaces are inductively defined from a finite set of constructors $T = \{\langle C_i, n_i, m_i \rangle\}_{i \in I}$, we can also "deconstruct" a represented problem $a \in TX$ in an inductive way. For instance, given a represented problem $a \in TX$ we can distinguish between the elements of $X$ which are "immediately accessible" and those which are part of some "sub-problem":

**Definition 3** (Immediate values and sub-problems). *Given a representation $T = \{\langle C_i, n_i, m_i \rangle\}_{i \in I}$ and a set $X$, define two functions* iv$: TX \to 2^X$ *(immediate values) and* sp$: TX \to 2^{TX}$ *(sub-problems), inductively as:*

- iv$(C_i \, x_1 \dots x_{n_i} \, a_1 \dots a_{m_i}) = \{x_1, \dots, x_{n_i}\}$
- sp$(C_i \, x_1 \dots x_{n_i} \, a_1 \dots a_{m_i}) = \{a_1, \dots, a_{m_i}\}$

The function iv extracts the immediate values contained in that represented problem (and ignores nested elements deeper down the representation), whereas sp performs the dual functionality, it gives us the set of sub-problems in a given representation, ignoring the immediate values. Using these we can define the *extension* of a represented problem $a \in TX$ as the subset $A \subseteq X$ whose elements are represented in $a$:

**Definition 4** (Extension map). *Given a representation $T = \{\langle C_i, n_i, m_i \rangle\}_{i \in I}$, define (uniformly in X) the extension map $[\![\cdot]\!] \in TX \to 2^X$ inductively as follows:*

$$[\![a]\!] = \mathrm{iv}(a) \cup \bigcup_{b \in \mathrm{sp}(a)} [\![b]\!]$$

This is well-defined on well-founded (i.e., finite) represented problems $a \in TX$. So the mapping $[\![\cdot]\!]$ translates represented decision problems into their set extension. For instance, if $X$ is the set of numbers then $T_{\mathsf{List}} X$ is the space of representations of lists of numbers. Using this representation, the list $[1, 2, 2]$ corresponds to the element $a = \mathsf{C}\,1\,(\mathsf{C}\,2\,(\mathsf{S}\,2))$ which has set extension $\{1, 2\}$

$$
\begin{aligned}
[\![\mathsf{C}\,1\,(\mathsf{C}\,1\,(\mathsf{S}\,2))]\!] \;&=\; \{1\} \cup [\![\mathsf{C}\,1\,(\mathsf{S}\,2)]\!] \\
&=\; \{1\} \cup \{1\} \cup [\![\mathsf{S}\,2]\!] \\
&=\; \{1\} \cup \{1\} \cup \{2\} \\
&=\; \{1, 2\}
\end{aligned}
$$

Hence, the extension map "extracts" the classical decision problem from a represented decision problem.

Another, hands on way to interpret the extension map is to view it from the perspective of an agent who wants to inspect all alternatives available in an online shop. In a such a shop, the agent might need to navigate through all the pages manually and inspect all the items by hand. The extension map instead would extract out all alternatives at once. (Comparable functionality exists for many web shops which allow users to show all results

in one page. Of course, even there, depending on the number of alternatives, agents will need to scroll down the page manually and will not perceive all elements at once).

**Definition 5** (Equivalent represented decision problems). *We say that two represented decision problems $a, b \in TX$ are extensionally equivalent, written $a \sim b$, if they represent the same set, i.e., $[\![a]\!] = [\![b]\!]$.*

Although $a = \mathsf{C}\,1\,(\mathsf{C}\,1\,(\mathsf{S}\,2))$ and $b = \mathsf{C}\,2\,(\mathsf{C}\,2\,(\mathsf{S}\,1))$ are different represented problems, they are extensionally equivalent as they both represent the set $\{1, 2\}$.

Conversely, we can also consider functions that take a classical decision problem $A \subseteq X$ and represent it as an element of the representation space $TX$. Hence, a represented decision problem can be thought of as an "enriched" classical decision problem.

**Definition 6** (*T*-representation map). *Let $2^X$ denote the set of non-empty finite subsets of X. A map $r \colon 2^X \to TX$ will be called a T-representation map if $A = [\![r(A)]\!]$, for any $A \subseteq X$.*

A *T*-representation map provides a specific way of representing a classical problem as an element of the representation space $TX$. We assume in this paper that the representations $TX$ admit a *T*-representation map. This is necessary so that any decision problem $A \subseteq X$ can be represented by an element $a \in TX$, i.e., $[\![a]\!] = A$. All of the standard representations such as lists and trees admit a representation map.

The *T*-representation map can also be practically interpreted from the perspective of a web shop. In essence, it describes how alternatives should be represented. Should they be listed? If so, how many items can a customer view at once? How will a customer proceed to the next results? Or, should alternatives be organized in a tree, for instance by a configuration assistant?

*3.3. Intensional Aspects of the Representation*

Let us conclude this section discussing purely *intensional* aspects of the representation, i.e., properties of the representation that are independent of the values being represented. For instance, given a represented problem $a \in TX$, we can navigate the representation in a depth-first search and inductively replace the values in $X$ by natural numbers (see Figure 2 where this is done for elements of $T_{\mathsf{List}}\,X$ with $X = \{a, b, c\}$).

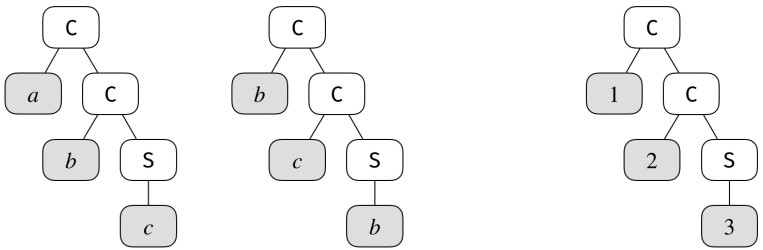

**Figure 2.** Two lists over $X = \{a, b, c\}$ (**left**) and their common index representation (**right**).

**Definition 7** (Index representation). *Given a representation $T = \{\langle \mathsf{C}_i, n_i, m_i \rangle\}_{i \in I}$ and a set X, let $\iota_X \colon TX \to T\mathbb{N}$ denote the function which, given a represented problem $a \in TX$, will inductively (in a depth-first search) replace the values in X by indices in $\mathbb{N}$, returning an element $\iota_X(a) \in T\mathbb{N}$. We call $\iota_X(a)$ the index representation of the represented problem $a \in TX$.*

We can associate with each occurrence of some $x \in X$ in $a \in TX$ its corresponding index, and also the value $x \in X$ of a given index $i \in \mathbb{N}$ in $a \in TX$. For example, given the list of names

$$a = \mathsf{C}\,\mathsf{Mary}\,(\mathsf{C}\,\mathsf{John}\,(\mathsf{S}\,\mathsf{Mary}))$$

its index representation is

$$\mathsf{C}\,1\,(\mathsf{C}\,2\,(\mathsf{S}\,3))$$

so we can say the last occurrence of "Mary" has index 3, and the value at index 2 is "John". We will write $a[i]$ for the value at index $i$ in the decision problem $a$, so, for instance, in the example above we have $a[2] = $ John and $a[3] = $ Mary.

Looking at the index representation is a powerful way of separating the structure of the representation from the concrete set being represented.

**Definition 8** (Equivalent underlying representation). *Given $a, b \in TX$, let*

$$a \simeq b \ \equiv \ \iota(a) = \iota(b)$$

*be the equivalence relation on $TX$ which identifies two represented problems that have the same underlying index representation.*

For instance, in Figure 2 the two lists on the left have the same underlying representation, which is captured by the index representation on the right.

**Definition 9** (Representation space quotient). *We will denote by $TX/\simeq$ the quotient space of $TX$ consisting of the equivalence classes of $\simeq$. We use $\mathbf{a}, \mathbf{b}$ for the elements of $TX/\simeq$. It is also easy to see that the mapping $\iota \colon TX \to T\mathbb{N}$ can be lifted to an injection $\iota^* \colon TX/\simeq \to T\mathbb{N}$.*

Indeed, by the definition of $\iota \colon TX \to T\mathbb{N}$ and the relation equivalence class $a \simeq b$, for each $\mathbf{a} \in TX/\simeq$ all the represented problems in $a \in \mathbf{a}$ will map to the same index representation $a^* = \iota(a) \in T\mathbb{N}$, so we have $\iota^*(\mathbf{a}) = a^*$.

## 4. Decision Procedures

In the previous section we described a general approach to deal with the *representation* of decision problems via the notion of a representation space $TX$ for the set of alternative $X$. Each decision problem $A \subseteq X$ may be represented in different ways as elements $a \in TX$. Similarly, we can consider different *implementations* of a choice function $c \colon 2^X \to X$ as what we call a *decision procedure*:

**Definition 10** (Decision procedure). *A program $P \colon TX \to X$ such that $P(a) \in [\![a]\!]$, for all $a \colon TX$, is called a decision procedure.*

Therefore, a decision procedure $P \colon TX \to X$ computes an element $x = Pa$ from a *represented* decision problem $a \in TX$. Procedures are defined by a case-distinction on each of the constructors of $TX$. For each constructor we need to state how the procedure will operate, and what the next step in the decision process will be. In this way the representation of a decision problem influences how an agent can choose. (Note, the references to "program" is not accidental. In computer science, a decision procedure corresponds to a functional program operating on an algebraic data type. In the context of a program, the case distinctions in the decision procedure are referred to as "pattern-matching").

**Example 11** (First element on list). *Let us illustrate this point with our representation of lists over some set of alternative $X$, i.e., $T_{\mathsf{List}} X$. Suppose the agent always chooses the first element of the list. His/her procedure $P \colon T_{\mathsf{List}} X \to X$ can be described as*

$$P(a) = \begin{cases} x & \text{if } a = \mathsf{S}\, x \\ x & \text{if } a = \mathsf{C}\, x\, xs \end{cases}$$

*Given $a = \mathsf{S}\, x$, a single element list, the procedure chooses this single element $x$. Given $a = \mathsf{C}\, x\, xs$, a compound list containing a "head" element $x$ and a "tail" $xs$, the procedure also chooses the head. Since any element of $T_{\mathsf{List}}$ is in either one of the two forms, the procedure is a well defined function.*

**Example 12** (Left-most element on binary tree). *Consider now a representation space for binary trees $T_{\mathsf{Tree}} = \{\langle \mathsf{N}, 1, 0\rangle, \langle \mathsf{B}, 0, 2\rangle\}$. So the elements of $T_{\mathsf{Tree}} X$ are binary tree representations of subsets of X. These will contain single "node" elements $\mathsf{N}\,x$, for values $x \in X$, and internal "branch" elements $\mathsf{B}\,a_l\,a_r$, where $a_l, a_r$ are the "left" and "right" sub-trees. In this case, a decision procedure $P\colon T_{\mathsf{Tree}} X \to X$ on binary trees that always selects the left-most element on the tree can be defined recursively as*

$$P(a) = \begin{cases} x & \text{if } a = \mathsf{N}\,x \\ P(a_l) & \text{if } a = \mathsf{B}\,a_l\,a_r \end{cases}$$

*Note how each of the two rules above describes a local decision: a decision on a final node, or a decision on an internal branch. A global choice process evolves from the iteration of these local descriptions via recursion.*

*Relation to Choice Functions*

In the same way as a decision problem can have multiple representations, a choice function can be implemented in different ways. Note, however, each choice function $c\colon 2^X \to X$ gives rise to one *canonical* decision procedure: Given a represented problem $a \in TX$, look at its set extension $A = [\![A]\!]$ and apply the choice function to this set $x = c([\![a]\!])$.

**Definition 13** (Canonical decision procedure from a choice function). *Given a choice function $c\colon 2^X \to X$, we can use the extension mapping $[\![\cdot]\!]\colon TX \to 2^X$ to lift $c\colon 2^X \to X$ into a decision procedure, which we denote by $[\![c]\!]\colon TX \to X$, as*

$$[\![c]\!](a) = c([\![a]\!])$$

*We will call $[\![c]\!]$ the canonical decision procedure associated with the choice function $c\colon 2^X \to X$.*

However, this is just one of the possible ways in which a choice function can be "implemented" as a decision procedure. Suppose the choice function is simply maximizing over some strict preference relation. When implementing this choice function as an actual procedure on the representation, one would have to decide on how to search the represented problem, which could be done for instance as a depth-first-search (starting from left or right) or a breadth-first-search. Although these will all lead to the same element being chosen (assuming a strict preference relation), the computational costs of the different search strategies may be very different for each concrete choice of representation.

Now, given a $T$-representation map $r\colon 2^X \to TX$, i.e., a fixed way of representing decision problems $A \subseteq X$, then we can also convert a decision procedure $P\colon TX \to X$ into a choice function: Given a decision problem $A \subseteq X$, first represent it as an element $r(A) \in TX$ and then apply the decision procedure $x = P(r(A))$.

**Definition 14** (Choice function from decision procedure). *Given a decision procedure $P\colon TX \to X$ and a $T$-representation map $r\colon 2^X \to TX$, define the choice function $\{\!\{P\}\!\}_r\colon 2^X \to X$ as*

$$\{\!\{P\}\!\}_r(A) = P(r(A))$$

## 5. Decision Procedures and Rational Choice

When is a *decision procedure* rationalizable by a strict preference relation? To answer this question, we begin by introducing some essential axioms of decision procedures.

*5.1. Properties of Decision Procedures*

The first property captures the notion of "divide-and-conquer":

**Axiom 15** (Inductive procedure). *A decision procedure $P: TX \to X$ is said to be inductive* (**IND**), *if the choice on a given decision problem is either one of the immediate values, or the same as a choice on one of the sub-problems: Formally, for all decision problems $a \in TX$*

$$P(a) \in \mathrm{iv}(a) \text{ or there exists } b \in \mathrm{sp}(a) \text{ such that } P(a) = P(b)$$

An agent whose procedure fulfills **IND** decides by dividing the overall problem and choosing on the sub-problems. The overall choice is determined by the choices on the sub-problems. For instance, in a restaurant, an agent who first decides whether to eat fish and then decides for sea bass would follow such a strategy.

A similar notion was introduced into decision theory by [8] under the name path-independence. There is an important difference though. Plott's property is defined in terms of choice functions, i.e., behavior. **IND**, in contrast, is defined for the decision procedure, i.e., the description of how agents proceed.

**IND** is also related to *partition independence*, a property introduced in [13] as their interpretation of Plott's path independence property. It is defined for choice functions selecting elements from lists. As Plott's property it is applied in terms of behavior. If we wanted to translate their property in terms of choice procedures, partition independence would be a considerably stronger notion than **IND** on lists. The reason is that, in combination with their definition of lists, in [13] agents have access to the whole list at once and can break it up at arbitrary positions. Partition independence implies that procedures are insensitive to where the list has been separated. The inductive property only implies agents apply the procedure "consistently" on the sub-problems. The latter are defined by the constructors of a representation. Thus, another way to view our results is that we can prove rationalizability for representations in general while imposing less structure than partition independence would.

The property **IND** focuses on how a decision procedure operates on a problem. Next we turn to the question how a procedure makes use of representation. There are three possibilities. Given a represented problem $a \in TX$, a procedure can

(1)   choose to ignore the representation completely, and choose only based on the set extension of the problem $[\![a]\!]$ (e.g., maximizing over some strict preference relation),
(2)   choose to ignore the elements themselves completely, and choose purely based on the representation (e.g., *first element on list*)
(3)   or a combination of the above.

The following property captures class (1):

**Axiom 16** (Extensional procedure). *A decision procedure is extensional* (**EXT**) *if it does not make use of the representation of the problem when making the decision, i.e., it produces the same outcome on different representations of the same decision problem:*

$$\forall a, b \in TX(a \sim b \;\Rightarrow\; Pa = Pb)$$

*Recall that $a \sim b$ is defined as $[\![a]\!] = [\![b]\!]$, i.e., a and b are representations of the same set.*

The next definition captures class (2):

**Axiom 17** (Intensional procedure). *A decision procedure is intensional* (**INT**) *if it does not make use of the actual values of the represented decision problem, but chooses simply based on the representation structure. We can make use of the quotient space $TX/\simeq$ to make this precise:*

$$\forall \mathbf{a} \in TX/\simeq \; \exists i \in \mathbb{N} \forall a \in \mathbf{a}(P(a) = a[i])$$

*Recall that we write $a[i]$ for the value $x \in X$ at index $i \in \mathbb{N}$ in the decision problem $a \in TX$.*

One should read the above as: Given a class of represented problems that only differ by the values, but not by the underlying representation, there is one particular position or index *i* which always contains the chosen element for all problems in that given class.

Hence, these two properties capture extreme classes of decision procedures: An extensional procedure strips away the representation and only considers the classical decision problem. That is, an agent who chooses in such a way needs access to all alternatives. If faced with having to navigate through all alternatives or getting just all alternatives at once, such an agent may prefer the latter.

A prominent example of an extensional procedure is of course a **maximizing** agent (Example 18):

**Example 18** (Maximizing). *Consider a procedure $P\colon TX \to X$ that recursively goes through a represented decision problem $a \in TX$ to find the maximal element according to some predetermined strict preference relation $(X, \succ)$, i.e., a complete, transitive and anti-symmetric relation. Such a procedure can be recursively defined as: For each constructor $\mathsf{C}_i$ of $TX$ (of arities $n_i$ and $m_i$)*

$$P(\mathsf{C}_i\, x_1 \dots x_{n_i}\, a_1 \dots a_{m_i}) = \max_{\succ}(\{x_1, \dots, x_{n_i}\} \cup \{P(a_j) \mid j \in \{1, \dots, m_i\}\})$$

*Note that this procedure can operate on different representations.*

An intensional procedure, in contrast, considers only the representation. Hence, the choice is independent of the concrete alternatives and dependent only on the representation. Examples of intensional procedures are the **first option on a list** (Example 11) or the **left-most option on a binary tree** (Example 12). (Examples A1, A2, and A4 which we describe in Appendix A belong to class (3) as they are neither fully extensional nor fully intensional).

Such examples may seem contrived at first. Why would anyone choose in such a way without considering the alternatives? However, consider an agent who searches on Google and just picks the first result. Or an agent who goes on Amazon and picks the first recommended alternative. Or an agent who in their favorite restaurant asks the waiter what wines he or she would recommend for a given meal and just goes with the first recommendation. We posit that such simple choice procedures are part of how people sometimes choose. Moreover, we posit that such procedures particularly make sense when the representations of alternatives and the choice environment more generally carries information that is meaningful for agents. We will revisit this aspect in Section 6.

*5.2. Rationalizable Choice Procedures*

We begin with the standard notion of a *choice function* which can be rationalized by a strict preference relation, i.e., such choices can be described as the outcome of maximizing a strict preference relation by an agent (see [6] Chapter 3).

**Definition 19** (Strictly rationalizable choice function). *A choice function $c\colon 2^X \to X$ is* strictly rationalizable *if there exists a strict preference relation $(X, \succ)$ such that for all decision problems $A \subseteq X$*

$$c(A) = \max_{\succ} A$$

*The strictness of the preference relation ensures that this maximum element is always unique.*

Given the canonical way of converting a choice function into a procedure (Definition 13), a procedure is strictly rationalizable if it has the same input-output behavior as the canonical procedure of a rationalizable choice function. Formally:

**Definition 20** (Strictly rationalizable procedure). *A decision procedure $P\colon TX \to X$ is said to be* strictly rationalizable *if $P = [\![c]\!]$, for some strictly rationalizable choice function $c\colon 2^X \to X$, i.e.,*

$$\forall a \in TX,\ P(a) = \max_{\succ}([\![a]\!])$$

*for some strict preference relation* $(X, \succ)$.

In other words, $P$ is strictly rationalizable if it "implements" a choice function whose choices agree with the maximization of a strict preference relation.

From now on, in order to simplify terminology, we will simply refer to "rationalizable choice" or a "rationalizable procedure". Throughout the paper we have in mind the rationalizability by a strict preference relation. Obviously, we could consider weaker forms of rationalizability, say allowing for indifference. However, the case of strict preferences is the most demanding and puts the most restrains on how an agent can use representations.

The following result shows that being extensional and inductive fully characterizes rationalizable procedures:

**Theorem 21.** *A procedure P is rationalizable iff P is* **EXT** *and* **IND**.

Thus, in order to choose rationally, a procedure needs to (i) ignore the representation completely and (ii) needs to proceed by a divide-and-conquer strategy. Not surprisingly, Example 18 (**maximization**) fulfills both properties.

The **satisficing** procedure (Example 22) illustrates a case where **IND** holds, but **EXT** does not.

**Example 22** ($T_{\mathsf{List}}$ Satisficing). *Consider the satisficing $T_{\mathsf{List}}$-decision procedure whereby an agent has an evaluation function $u \colon X \to \mathbb{R}$ and a satisficing (This example is based on [6], p. 30). threshold $u^* \in \mathbb{R}$. The agent chooses the first element of the list that is above the given threshold. If there is none, he or she chooses the last element. This procedure can be recursively defined as:*

$$P(a) = \begin{cases} x & \text{if } a = \mathsf{S}\, x \\ x & \text{if } a = \mathsf{C}\, x\, a' \text{ and } u(x) \geq u^* \\ P(a') & \text{if } a = \mathsf{C}\, x\, a' \text{ and } u(x) < u^* \end{cases}$$

It is easy to see that **satisficing** is an inductive procedure. However, it does not fulfill **EXT**: consider two lists $a_1, a_2$ which contain the same elements $\{x, y\}$, i.e., $[\![a_1]\!] = [\![a_2]\!] = \{x, y\}$, but suppose $x$ comes first in $a_1$, while $y$ comes first in $a_2$. Suppose also that $u(x), u(y) > u^*$. Then, $P(a_1) = x$ and $P(a_2) = y$. Thus, $P$ is not extensional and hence is not rationalizable.

*5.3. Discussion*

Overall, Theorem 21 has strong implications. If rational agents must ignore the representation of the problem completely, does that mean that representations are meaningless? To date, we only required that representations give the decision problem some structure. Apart from that we did not impose other restrictions.

Yet, representations are rarely arbitrary. There is typically some logic behind. The pages on a search engine are listed according to some "page rank", the items on a restaurant menu are organized according to categories (starter, main, dessert), the products in an online shop are displayed either by relevance, average customer rating or price, etc.

In the next section, we explore the idea that, when restricted to a "meaningful" set of representations, it is indeed possible for agents to choose in a purely intensional way (e.g., always choosing the first hit on a search engine) while still being rational (e.g., maximizing page rank).

## 6. Generalization: Meaningful Representations

In this section, we consider representations which carry additional structure. That is, instead of considering the complete representation space $TX$, we might wish to restrict ourselves to some subset of "valid" or "meaningful" representations $\mathcal{V} \subseteq TX$. For instance, we may be interested in lists where the order in which the elements are listed is not

arbitrary but follows some rule. Nevertheless, we want that any decision problem is still representable as an element of $\mathcal{V}$:

**Definition 23** (Expressive $\mathcal{V}$). *We call $\mathcal{V}$ expressive if for any subset $A \subseteq X$ there exists an $a \in \mathcal{V}$ such that $\llbracket a \rrbracket = A$. We say that $\mathcal{V}$ is* uniquely expressive *if such $a \in \mathcal{V}$ is uniquely determined by $A$.*

All properties of decision procedures introduced above relativize to subsets of $TX$. For instance, we can define the notion of rationalizability (Definition 20) relative to a set of valid representations $\mathcal{V} \subseteq TX$:

**Definition 24** ($\mathcal{V}$-rationalizable procedure). *Let $\mathcal{V} \subseteq TX$. A decision procedure $P\colon TX \to X$ is said to be $\mathcal{V}$-rationalizable if there exists a rationalizable choice function $c\colon 2^X \to X$ such that*

$$\forall a \in \mathcal{V}, \; P(a) = c(\llbracket a \rrbracket)$$

We can also relativize the notions of **EXT** and **IND** to represented problems in $\mathcal{V}$, and we call these $\mathcal{V}$-**EXT** and $\mathcal{V}$-**IND**. Furthermore, one direction of Theorem 21 relativizes to any expressive set $\mathcal{V} \subseteq TX$, i.e.,

**Theorem 25.** *Assume $\mathcal{V}$ is expressive. If a procedure $P$ is $\mathcal{V}$-rationalizable, then it is $\mathcal{V}$-**EXT** and $\mathcal{V}$-**IND**.*

For the converse direction, however, we need to impose a further requirement on the set of valid representations $\mathcal{V}$:

**Definition 26** ($T$-closed $\mathcal{V}$). *Let $T = \{\langle \mathsf{C}_i, n_i, m_i \rangle\}_{i \in I}$. We say that $\mathcal{V}$ is $T$-closed if whenever $\{a_1, \ldots, a_{m_i}\} \subseteq \mathcal{V}$ then $\mathsf{C}_i\, x_1 \ldots x_{n_i}\, a_1 \ldots a_{m_i} \in \mathcal{V}$, for any $\{x_1, \ldots, x_{n_i}\} \subseteq X$.*

**Theorem 27.** *Assume $\mathcal{V}$ is expressive and $T$-closed. If a procedure $P$ is $\mathcal{V}$-**EXT** and $\mathcal{V}$-**IND** then it is also $\mathcal{V}$-rationalizable.*

We concluded the previous section by arguing that the **satisficing** procedure of Example 22 is not rationalizable because it is not extensional. Consider, however, the following variant of satisficing where the set of representations is restricted:

**Example 28** (Satisficing with restricted representations). *Let $(X, <)$ be a totally ordered set. Let $\mathcal{V} \subseteq T_{\mathsf{List}} X$ consist only of lists where the elements are arranged according to the order $<$, so that if $a \in \mathcal{V}$ and $i < j$ then $a[i] \leq a[j]$. We can consider now the same procedure as in Example 22, but now on this restricted set of lists.*

This restricted set of lists $\mathcal{V} \subseteq T_{\mathsf{List}} X$ is both expressive and $T$-closed. The restriction on representations also ensures that this procedure, although not globally extensional, is extensional on the restricted domain of lists that satisfy the global ordering $(X, <)$. Moreover, as we argued before, the satisficing procedure is inductive. Therefore, by Theorem 27, the restricted satisficing procedure is $\mathcal{V}$-rationalizable. This fact, that the above version of satisficing can be rationalized if constrains are put on the list representation, is discussed in [6], p. 30.

It is easy to see that for any decision problem $A \subseteq X$ there is only one way of representing $A$ as a list $a \in T_{\mathsf{List}} X$ if one at the same time insists that the elements are listed according to some total ordering $(X, <)$. In general, if decision problems have a unique representation in $\mathcal{V}$, then any procedure is trivially $\mathcal{V}$-**EXT** (there are no two problems with different representations). The following theorem shows that the converse must also hold:

**Theorem 29.** *For any expressive $\mathcal{V}$, it is uniquely expressive iff all procedures $P$ are $\mathcal{V}$-**EXT**.*

As a consequence, if all procedures—when restricted to $\mathcal{V}$—are either rational or extensional, then $\mathcal{V}$ must be uniquely expressive:

**Corollary 30.** *If every procedure P on TX is either $\mathcal{V}$-rational or $\mathcal{V}$-**EXT**, then $\mathcal{V}$ must be uniquely expressive.*

Finally, we conclude this paper by considering procedures which are *purely intensional* and $\mathcal{V}$-*rational*. We prove that in such cases the set $\mathcal{V}$ must fix an index $i$ for all problems in $\mathcal{V}$ with the same structure and the same value:

**Theorem 31.** *For any expressive $\mathcal{V}$, if $P\colon TX \to X$ is **INT** and $\mathcal{V}$-rational then*

$$\forall \mathbf{a} \in TX/\simeq \ \exists i^{\mathbb{N}} \forall A \subseteq X \exists v^X \forall a \in \mathcal{V} \cap \mathbf{a}(\llbracket a \rrbracket = A \to a[i] = v)$$

Indeed, if an agent is choosing rationally, their choice has to be the same on all $a \in \mathcal{V}$ which are representations of the same set $A \subseteq X$. But if their procedure is also purely intensional, the agent is choosing by always selecting a particular index on the represented problem, irrespective of the values being represented. Such a combination is only possible when the set $\mathcal{V}$ ensures that for all $a \in \mathcal{V}$, which represent the same set $A \subseteq X$, a particular index $i \in \mathbb{N}$ always contains the same value $v \in X$—which is the maximal value on some strict preference relation on $X$. Nothing needs to be said about all other indices $j \neq i$ though.

Let us entertain the following thought experiment: Suppose a search engine comes up with a ranking of pages which perfectly matches the preferences of the users over web pages. The search engine then lists pages, for each search term, in decreasing order of page rank. The users, once aware of this, will always pick the first page on the search result (a purely intensional procedure). This is only rational because the "best" page is always placed on the first position. However, that, paradoxically, also implies that it does not really matter what pages come at 2nd, 3rd, ... positions, as these are never chosen by this **first element on list** procedure.

## 7. Conclusions

If customers shop online, they do not interact with the set of alternatives directly but with a representation of these alternatives. In this paper, we introduce a novel framework in which we can model these representation as well as the decision-making procedures of agents. We show how procedures induce *choice functions* which are well established in the economic literature.

In the context of choice functions, *rationalizability* is a key aspect. We thus take this as a starting point for our analysis and characterize under which conditions procedures operating on represented problems can be rationalized. Importantly, with adequate guarantees on representations the agent can choose in a purely intensional way, i.e., he or she can choose on the basis of representations alone.

Two immediate implications follow from our results. First, through the mapping from internal procedures to choice functions, we can "lift" rationality criteria from choice functions to procedures. What this means: If an agent describes how he or she is actually choosing, we can analyze that description in our language and directly tell—without a single choice being made—whether the agent is maximizing a rational preference relation or not. Moreover, in principle, we can apply a similar strategy to click stream data (as for instance collected by online shops). Our results could serve as the basis for new tests of rationality assumptions not by comparing different choices—as is usual in the empirical literature on revealed preferences—but instead by tracking the choice process. The same methodology may be applicable to other theories of decision-making.

Secondly, the ability to integrate observable and measurable data structures into a model can not only be used to extend tests for existing theories but it can also be used to develop new models of decision-making. Today, there is a big gap between classical choice

theory which operates on rudimentary data such as sets of alternatives and choices and the actual observable data. The question is how can one integrate such data with classical choice theory and develop new models? The representations of alternatives we provide is a way to do so.

We noted already that the formal treatment we use here is standard in theoretical computer science and functional programming. It is a formal language to describe representations (algebraic data types) and procedures (programs) operating on these representations, while being a formal language, which one can use for mathematical reasoning as we do here in the paper, this language can also be implemented in modern functional programming languages.

Consider again the representation of a list: $T_{\mathsf{List}} = \{\langle \mathsf{S}, 1, 0 \rangle, \langle \mathsf{C}, 1, 1 \rangle\}$ and the example list $a = \mathsf{C}\,1\,(\mathsf{C}\,2\,(\mathsf{S}\,2))$

We can implement this directly in a functional programming language such as Haskell (https://www.haskell.org/, (accessed on 24 October 2021)). as follows:

```
(1) data TList x = S x | C x (TList x)
```

```
(2) a = C 1 (C 2 (S 2))
```

(1) gives a definition of $T_{\mathsf{List}}$ (known in Haskell as a data type). Note, that as in our formal definition the Haskell data type is parametric with respect to the exact object it shall represent (and is therefore parameterized by the variable "$x$"). (2) is the representation of the list $[1, 2, 2]$ as an instance of that data type. The similarity of the Haskell representation to the formal representation is not a coincidence. The formal structure of the representations we have introduced can be directly implemented in modern programming languages which support this functionality (as in the example above for instance in Haskell). As a consequence of this, all representations we introduce here can be directly implemented, and be used for simulation of procedures, for instance. Or one could define a representation framework (like an online shop), translate it into the formal framework to reason about it, or match it to data.

Regarding future work, the framework we have introduced provides an interface to a whole new toolkit that can be applied to choice scenarios. We will sketch two of these.

First, in this paper, we follow the notion of behavior as it is standard textbooks: a choice of an alternative. However, online shops do gather a lot more information about customers than just a final choice. In fact, in many situations customers may not make a buy decision in the end. Still, their navigation through a web shop may reveal relevant information about their decision-making: Which elements do they inspect? Where do they spend their time? Nowadays, such information is routinely collected and analyzed. The methodology we have introduced here can handle different forms of such "behavior". Recall the type of procedure we consider:

$$P \colon TX \to X$$

We can augment the outcome type to:

$$P \colon TX \to MX$$

where $M$ forms some algebraic structure on top of the set of alternatives. Concretely, this could be a non buy-decision; it could be a probability distribution, i.e., we behavior is not deterministic but probabilistic; it could also mean that we are not tracking the final choice but the intermediate steps—as often measured in clickstream data. In the theoretical computer science literature the structure of $M$ is well studied. Furthermore, the cases like probability, or "exceptions" or "trace" can be directly interpreted as the aforementioned choice theoretic terms.

A second extension concerns the complexity of procedures, which we do not touch in the current paper. Yet, our paper does set up a key prerequisite for such an analysis:

the distinction between procedure and behavior. Why? Because it allows us to define the equivalence of procedures (as resulting in the exact same behavior). Equipped with the notion of equivalence, we can cleanly compare procedures.

This is central if we want to understand the value of different representations. Consider the following example: You want to buy an external harddrive. You go to an online platform and search for the available alternatives. Suppose you have clear preferences about attributes of the harddrive, for instance what type of disk, capacity, noise level etc. Suppose further that the platform offers decision support: you can filter and sort alternatives according to some attributes. *How* you can find the best alternative depends on that decision support. Designed in an adequate way, it can lead you quickly to your best choice. Designed in an inadequate way (for instance by not providing the functionality to filter out alternatives), you might need to inspect alternatives step by step.

Indeed, in another paper [29], we show under which conditions such filtering/sorting actions are adequately designed and will be "quicker" than maximization. Of course, this issue is by no means limited to maximization. The representation of alternatives and the information available in general affects how people can choose. Here, we introduce a framework which can be extended in these directions.

**Author Contributions:** Both authors have contributed equally to the manuscript. All authors have read and agreed to the published version of the manuscript.

**Funding:** This research received no external funding.

**Institutional Review Board Statement:** Not applicable.

**Informed Consent Statement:** Not applicable.

**Data Availability Statement:** Not applicable.

**Acknowledgments:** We thank Thomas Epper, three anonymous referees, and seminar audiences at the University of St. Gallen and the European Meeting of the Econometric Society 2019 for helpful comments.

**Conflicts of Interest:** The authors declare no conflict of interest.

## Appendix A. Additional Examples

In this section, we provide additional examples which are neither fully extensional nor fully intensional.

**Example A1** ($T_{\mathsf{Tree}}$ Satisficing)**.** *Consider also a* satisficing *procedure on a binary tree, $T_{\mathsf{Tree}}\,X$, whereby an agent has an evaluation function $u\colon X \to \mathbb{R}$ and a satisficing threshold $u^* \in \mathbb{R}$, and the agent chooses the left-most element of the binary tree that is above the given threshold. If there is none, he or she chooses the right-most element. This procedure can be recursively defined as:*

$$P(a) = \begin{cases} x & \text{if } a = \mathsf{N}\,x \\ P(a_l) & \text{if } a = \mathsf{B}\,a_l\,a_r \text{ and } u(P(a_l)) \geq u^* \\ P(a_r) & \text{if } a = \mathsf{B}\,a_l\,a_r \text{ and } u(P(a_l)) < u^* \end{cases}$$

**Example A2** (Conditional $T_{\mathsf{Tree}}$ satisficing)**.** *The set of alternatives $X$ is partitioned into two sets, $K$ (known, or default) and $N$ (new). Given a problem $a \in T_{\mathsf{Tree}}\,X$, if more than half of the elements in $[\![a]\!]$ are known elements, choose the first (on a depth-first search) element of the binary tree which is in $K$. Otherwise choose the maximal element of the binary tree according to some predetermined order $(X, \succ)$ as in Example 18.*

**Example A3** (Choose default on large problems)**.** *Assume the agent has a default choice $x \in X$ in mind and a size threshold $N$. Consider the procedure that given $a \in TX$ returns $x$ when $x \in [\![a]\!]$ and $|[\![a]\!]| > N$, or else returns the maximal element in $[\![a]\!]$ according to some predetermined order $(X, \succ)$ as in Example 18.*

**Example A4** (Avoiding undesirable element). *Consider a procedure for selecting an element from a binary tree of alternatives $a \in T_{\mathsf{Tree}} X$. The procedure will recursively operate on the left sub-tree and take that choice, except when that choice is an undesirable element $x^* \in X$ and the number of choices is large (bigger than some given threshold $N \geq 1$), in which case it disregards the left choice and works recursively on the right sub-tree. The element $x^*$ and the threshold $N$ are fixed exogenously. The procedure can be defined as:*

$$P(a) = \begin{cases} x & \text{if } a = \mathsf{N}\, x \\ P(a_l) & \text{if } a = \mathsf{B}\, a_l\, a_r \text{ and } P(a_l) \neq x^* \text{ or } |[\![a_l]\!] \cup [\![a_r]\!]| \leq N \\ P(a_r) & \text{if } a = \mathsf{B}\, a_l\, a_r \text{ and } P(a_l) = x^* \text{ and } |[\![a_l]\!] \cup [\![a_r]\!]| > N \end{cases}$$

**Example A5** (Larger sub-problem bias). *Consider again an agent choosing on binary trees $a \in T_{\mathsf{Tree}} X$, but this time suppose they also have a size threshold $N \geq 1$. The agent always recursively chooses to work on the biggest of the two sub-trees, and maximizes when reaching a sub-problem of size smaller or equal to $N$:*

$$P(a) = \begin{cases} \max_{\succ} [\![a]\!] & \text{if } |[\![a]\!]| \leq N \quad (\text{as in Example } 18) \\ P(a_1) & \text{if } a = \mathsf{B}\, a_1\, a_2 \wedge |[\![a_1]\!]| > |[\![a_2]\!]| \\ P(a_2) & \text{if } a = \mathsf{B}\, a_1\, a_2 \wedge |[\![a_1]\!]| \leq |[\![a_2]\!]| \end{cases}$$

## Appendix B. Proofs

Proof of Theorem 21:

**Proof.** First, assume $P$ is rationalizable. Let $c \colon 2^X \to X$ be a rationalizable choice function such that

(i)  $P(a) = c([\![a]\!])$, for all $a \in TX$.

Let $(X, \succ)$ be the *strict preference relation* underpinning $c$, i.e.,

(ii)  $c(A) = \max_{\succ} A$

In order to show **EXT**, let $a, b \in TX$ be two represented problems such that $[\![a]\!] = [\![b]\!]$. Clearly, $c([\![a]\!]) = c([\![b]\!])$, and hence, by (i), $P(a) = P(b)$. In order to show that $P$ is also **IND**, let $a \in TX$ be a fixed represented problem, and assume $P(a) \notin \mathrm{iv}(a)$. Since $P(a) \in [\![a]\!]$ there must be a $b \in \mathrm{sp}(a)$ such that $P(a) \in [\![b]\!]$. By (ii), $P(a)$ is a maximal element in $[\![a]\!]$, and hence also in $[\![b]\!]$, since $[\![b]\!] \subseteq [\![a]\!]$. Therefore, $P(b) = P(a)$.

For the converse, assume $P$ is both **EXT** and **IND**. Define the following relation $x \succ_P y$

$$x \succ_P y \equiv \forall a \in TX (x, y \in [\![a]\!] \to P(a) \neq y)$$

In particular, this implies that $x$ is chosen in $a \in TX$ when $[\![a]\!] = \{x, y\}$. Let us first show that this relation is actually a preference relation. In order to see that it is *transitive*, assume $x \succ_P y$ and $y \succ_P z$, and fix $a \in TX$ such that $x, z \in [\![a]\!]$. We need to show that $P(a) \neq z$. Let $a'$ be any problem which has $a$ as a sub-problem, but any other sub-problem or immediate value only has $y$. By $y \succ_P z$, we have that $P(a') \neq z$. However, by $x \succ_P y$, since $x, z \in [\![a']\!]$, we also have that $P(a') \neq y$. However, since $P$ is inductive, and by the way $a'$ was constructed, we must have that $P(a') = P(a)$. Therefore, $P(a) \neq z$. In order to show that this relation is also *total*, suppose there were $x, y$ such that $\neg(x \succ_P y)$ and $\neg(y \succ_P x)$. That implies that for some $a$ and $b$, both containing $x$ and $y$, we have that $P(a) = x$ and $P(b) = y$. Let $C = [\![a]\!] \cap [\![b]\!]$, which is guaranteed to be non-empty since both $a$ and $b$ contain $x$ and $y$. Let $a'$ be such that $[\![a']\!] = [\![a]\!]$ but $a'$ has a sub-problem $c$ such that $[\![c]\!] = C$, and the only place where $x, y$ appear in $a'$ is in the sub-problem $c$. Similarly, we can also find a $b'$ such that $[\![b']\!] = [\![b]\!]$ but $b'$ also has $c$ as a sub-problem, with the same property that $x, y$ only appear in $c$. Since $P$ is extensional, we have that $P(a) = P(a')$ and $P(b) = P(b')$. However, since $P$ is also inductive, we must have that $P(a') = P(c)$ and $P(b') = P(c)$,

which is a contradiction since $P(a) = x \neq y = P(b)$. That concludes the proof that $x \succ_P y$ is a preference relation. It remains to see that $P(a) = \max_{\succ_P} [\![a]\!]$. However, that is indeed the case since $x \succ_P y$ implies that $y$ is never chosen when $x$ is present. If $y = P(a)$ was not the maximal element of $[\![a]\!]$ according to this order, there would be a $x \in [\![a]\!]$ with $x \succ_P y$, meaning that $y$ could not have been chosen in $a$, contradiction.  □

Proof of Theorem 25:

**Proof.** A simple relativization of the proof of Theorem 21. Assume $P$ is $\mathcal{V}$-rationalizable and let $c \colon 2^X \to X$ be a rationalizable choice function such that

(i)    $P(a) = c([\![a]\!])$, for all $a \in \mathcal{V}$.

Let $(X, \succ)$ be the *strict preference relation* underpinning $c$, i.e.,

(ii)   $c(A) = \{x \in A \mid \forall y \in A(x \neq y \to x \succ y)\}$.

In order to show **EXT**, let $a, b \in \mathcal{V}$ be two represented problems such that $[\![a]\!] = [\![b]\!]$. Clearly, $c([\![a]\!]) = c([\![b]\!])$, and hence, by (i), $P(a) = P(b)$. In order to show that $P$ is also **IND**, let $a \in \mathcal{V}$ be a fixed represented problem, and assume $P(a) \notin \mathrm{iv}(a)$. Since $P(a) \in [\![a]\!]$ there must be a $b \in \mathrm{sp}(a)$ such that $P(a) \in [\![b]\!]$. By (ii), $P(a)$ is a maximal element in $[\![a]\!]$, and hence also in $[\![b]\!]$, since $[\![b]\!] \subseteq [\![a]\!]$. Therefore, $P(b) = P(a)$.  □

Proof of Theorem 27:

**Proof.** Again, a simple relativization of the proof of Theorem 21. Let $\mathcal{V}$ be expressive and $T$-closed, and assume $P$ is both $\mathcal{V}$-**EXT** and $\mathcal{V}$-**IND**. Define the following relation $x \succ_P y$

$$x \succ_P y \equiv \forall a \in \mathcal{V}(x, y \in [\![a]\!] \to P(a) \neq y)$$

Let us first show that this relation is actually a preference relation. In order to see that it is *transitive*, assume $x \succ_P y$ and $y \succ_P z$, and fix $a \in \mathcal{V}$ such that $x, z \in [\![a]\!]$. We need to show that $P(a) \neq z$. Let $a' \in \mathcal{V}$ be any problem which has $a$ as a sub-problem, but any other sub-problem or immediate value only has $y$. Such $a'$ exists under the assumption that $\mathcal{V}$ is $T$-closed. By $y \succ_P z$, we have that $P(a') \neq z$. However, by $x \succ_P y$, since $x, z \in [\![a']\!]$, we also have that $P(a') \neq y$. However, since $P$ is inductive, and by the way $a'$ was constructed, we must have that $P(a') = P(a)$. Therefore, $P(a) \neq z$. In order to show that this relation is also *total*, suppose there were $x, y$ such that $\neg(x \succ_P y)$ and $\neg(y \succ_P x)$. That implies that for some $a, b \in \mathcal{V}$, both containing $x$ and $y$, we have that $P(a) = x$ and $P(b) = y$. Let $C = [\![a]\!] \cap [\![b]\!]$, which is guaranteed to be non-empty since both $a$ and $b$ contain $x$ and $y$. Let $a'$ be such that $[\![a']\!] = [\![a]\!]$ but $a'$ has a sub-problem $c$ such that $[\![c]\!] = C$, and the only place where $x, y$ appear in $a'$ is in the sub-problem $c$. Such $c'$ exists under the assumption that $\mathcal{V}$ is expressive ($c \in \mathcal{V}$ exists with $[\![c]\!] = C$) and $T$-closed ($c' \in \mathcal{V}$ exists with $c$ as a sub-problem). Similarly, we can also find a $b'$ such that $[\![b']\!] = [\![b]\!]$ but $b'$ also has $c$ as a sub-problem, with the same property that $x, y$ only appear in $c$. Since $P$ is extensional, we have that $P(a) = P(a')$ and $P(b) = P(b')$. However, since $P$ is also inductive, we must have that $P(a') = P(c)$ and $P(b') = P(c)$, which is a contradiction since $P(a) = x \neq y = P(b)$. That concludes the proof that $x \succ_P y$ is a preference relation. It remains to see that $P(a) = \max_{\succ_P} [\![a]\!]$. However, that is indeed the case since $x \succ_P y$ implies that $y$ is never chosen when $x$ is present. If $y = P(a)$ was not the maximal element of $[\![a]\!]$ according to this order, there would be a $x \in [\![a]\!]$ with $x \succ_P y$, meaning that $y$ could not have been chosen in $a$, contradiction.  □

Proof of Theorem 29:

**Proof.** From left-to-right this is trivial since when $\mathcal{V}$ uniquely determines the representation of a set $A \subseteq X$, then any two $a_1, a_2 \in TX$ such that $[\![a_1]\!] = [\![a_2]\!]$ will be such that $a_1 = a_2$, and hence $P(a_1) = P(a_2)$. For the other direction, suppose all procedures $P$ are $\mathcal{V}$-**EXT**. Assume $\mathcal{V}$ is not uniquely expressive. There must exist $a_1, a_2 \in TX$ such that $A = [\![a_1]\!] = [\![a_2]\!]$ but $a_1 \neq a_2$. Let $x, y$ be two distinct elements of $A$, and define a procedure $P$ such that

$P(a_1) = x$ but $P(a_2) = y$. This is always possible because procedures are able to inspect not just the values of $a_i$ but the structure of the representation. However, this is a contradiction because $P$ is by assumption $\mathcal{V}$-**EXT**. $\square$

Proof of Corollary 30:

**Proof.** Assume every $P$ on $TX$ is either $\mathcal{V}$-rational or $\mathcal{V}$-**EXT**, but suppose $\mathcal{V}$ is not uniquely expressive. By Theorem 29 there exists a procedure $P\colon TX \to X$ which is not $\mathcal{V}$-**EXT**. However, by Theorem 25, $P$ is also not $\mathcal{V}$-rationalizable, which is a contradiction. $\square$

Proof of Theorem 31:

**Proof.** Since $P\colon TX \to X$ is intensional, we know that for any given equivalence class $\mathbf{a} \in TX/\simeq$ the procedure $P$ always chooses the values on the same index $i$, i.e., $P(a) = a[i]$ for all $a \in \mathbf{a}$. Let $A \subseteq X$ be fixed and $a \in \mathcal{V}$ be such that $[\![a]\!] = A$. Such $a \in \mathcal{V}$ exists since we assume $\mathcal{V}$ is expressive. Let $v = P(a)$. We claim that for any other $b \in \mathcal{V} \cap \mathbf{a}$ with $[\![b]\!] = A$ we also have $b[i] = v$. Indeed, if $b \in \mathcal{V} \cap \mathbf{a}$ we have that $P(b) = b[i]$, but also $P(a) = a[i]$. By the assumption that $P$ is $\mathcal{V}$-rational, we must have that $b[i] = a[i] = \max_{\succ} A$, for some strict preference relation $(X, \succ)$, which implies $P(b) = b[i] = a[i] = v$. $\square$

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
