# Peer review of "On Rational Choice and the Representation of Decision Problems"

_games, doi:10.3390/g12040086_

Round 1

Reviewer 1 Report

The paper is interesting, but some changes should be done before considering it for publication. Amongst main relevant issues I suggest considering the following aspects:

In the abstract I miss some interesting practical implications. What are the theoretical. economic, social implications of this article?

In the introduction, before point 1.1. I miss after presenting the objective of research, to briefly describe the different parts of the article, and what it will be included in each of them.

Authors include it in the part 1.2. outline of the paper. I think it should be located before the related literature (point 1.1.)

At the beginning of part 2 I miss citations specially to support authors explaining that decision problem is modelled as a subset A of a set of alternatives or supporting the following facts: “The modelling of a decision problem as a set abstracts from how the problem is presented. “The modelling of the agent’s behaviour as a function also abstracts from the actual process of choosing.”

“In practice, however, agents do not see the set of alternatives (at once), but rather a representation of that set”

They are all facts previously contemplated by different researchers.

In part 2.1, I miss citations supporting the two properties referred to representations.

In parts 2.2 The extension and representation maps and in parts 4.1 and 4.2, I miss real life examples of the various aspects referred to.

In conclusion, practical implications of the article should be provided..

Reviewer 2 Report

Dear authors

I am recommending the manuscript to be considered after major revisions. The reason for this is that I am missing the practical aspect of your work. Given the journal you submitted to, I expect the manuscript to also address the question of what problem this solves and how this can be used (and validated). You yourself state that you omit any consideration regarding the complexity. I'd like to see at lest some elaboration on where these complexities might fall and what that means for the practical use of your work.

You state (on page 6) that "The challenge in this paper is setting up the framework, i.e. adequate definitions". You do that in Section 2 and 3. You then, in the conclusion, state that "we provide a model of representations which agents encounter when choosing among
alternatives" to determine / identify "under which conditions" agents' choices can be rationalized. What I am missing is why the reader would need to know this. You submitted the manuscript to the journal Games and I am not seeing the direct connection. I appreciate the theoretical work and the practical considerations but I am wondering whether this is the wrong venue for the manuscript. I do not think there is something wrong with the manuscript per se, I am just not sure what the benefit to the readers of the journal will be. This could be addressed by extending the conclusion to cast the benefits for practical implementations.

On page 3 you write "we want to make sure that the software makes good decisions before we let it loose". You mean "rational" decisions I assume, as goodness is determined by e.g., utility or some other objective function which is given to the system. Adherence to this value function IS how choices are made, correct? The question therefore is whether the choices made (as compared on the basis of their utility, or goodness) are rational.

Also on page 3 you state that the case where nothing is known is an important edge case, but you then discontinue the discussion and do not, in fact, discuss this "important" edge case at all. This is an example of how the theoretical work, while interesting, seems to lack the connection to context and practical considerations.

I noted (on page 4 Please check -- I am not 100% sure myself!) a potential miss-referenced term: I believe the term "satisficing" used on page 4 should reference to Simon 1956 ("Rational choice and the structure of the environment") and not Simon 1955 ("A Behavioral Model of Rational Choice").

A concluding comment: the paper is written in a light hearted, and sometimes colloquial, fashion. This is dangerous in the sense that it may attract criticism from reviewers. Given the depth of formalisms in the manuscript, I myself appreciate the fact that a good part of the paper can be read at ease. I would, however, not be surprised if this tone was perceived negatively by my fellow reviewers. In line with the previous sentiment, I would like to emphasize that I have not examined all formal statements for correctness and typos; I have not verified the proofs in the appendix myself. I hope that my fellow, more picky, reviewers have.

Reviewer 3 Report

Referee's report on manuscript number: games-1421463

On Rational Choice and the Representation of Decision Problems

Decision

Although this is a well-written paper and does not exhibit big mistakes, I feel that this paper needs to be changed. I believe that the contribution is innovative but some important changes must be made.

Comments

My opinion stands on the following considerations:

1. Introduction section should be changed. Now it is very strong. There is no motivation and authors introduce some concepts that are not introduced anywhere. Authors must explain in detail the type of problems that they want to represent. I think that include some illustrative examples in Introduction could help.

Moreover, authors do not justify why they select and representation of the alternatives by mean a tree. It is not clear enough. I feel that this kind of decision making problems are usual in some special scenarios but they are not general problems.

From another point of view, authors must explain if the agents know all alternatives. That is, If agents do not know all alternatives and they must to select one (that includes others), There are any risk associated to the decision that does not appear.

I understand that authors try to follow the contribution of Apesteguia and Ballester (2013) but their interpretation is not the same.

From my point of view, the authors do not introduce a new representation of a decision problem but they present a kind of algorithm.

2. Subsection 1.2 should be deleted. From my point of view, it should be included in Introduction but not like a subsection.

3. Subsection 1.1 should appear like a Section.

4. Section 2 must be improved. Notation is not clear enough if you are not an expert. Authors must indicate along the paper if the proofs are included on Appendix.

5. Conclusions are not clear enough. Authors must improve it.

Round 2

Reviewer 1 Report

I think authors have considered all my suggestions and recommendations and have included all the answers to my questions in the new version of the manuscipt. However, in conclusions there is a sentence that should be revised. It is this one, at the end of the conclusions: "Indeed in another paper, we show under which conditions such as filtering/sorting actions are adequately designed and will be “quicker” than maximization"

Do authors refer to a previous article that they have already published? In this case, they should include the citation of the article in the text in the conclusion. Or Do authors refer to a future manuscript they will do? In this case, authors should change the sentence for this one: 

"Indeed in another paper, we will show under which conditions such as filtering/sorting actions are adequately designed and will be “quicker” than maximization"

Reviewer 2 Report

Dear authors

Thank you for addressing my concerns in your response. It is a pleasure to see that my concerns, albeit not all addressed, have been responded to. I would like to open with the admission that I was not aware of the special issue to which this manuscript was submitted. This is probably my own oversight, but be that as it may, this addresses some of my concerns regarding the theoretical nature of your work. I have read your response and I acknowledge the your rebuttal.

In the closing paragraph of the conclusion, you mention the paper (which I assume is the companion paper you refer to in your response to my comments). Please reference the paper here; if it is not yet available consider placing it on arXiv (or some other repository) and referencing this instead. 

I still have my concerns about the purely theoretical nature of the paper but I concede. I have no further objections to publication of the manuscript. 

Reviewer 3 Report

There are not comments 

Author Response

No comments received - thanks again for your previous comments!